# Design, Development and Evaluation of Thermal Properties of Polysulphone–CNT/GNP Nanocomposites

**DOI:** 10.3390/nano11082080

**Published:** 2021-08-16

**Authors:** Hafiz Muzammil Irshad, Abbas Saeed Hakeem, Kabeer Raza, Turki Nabieh Baroud, Muhammad Ali Ehsan, Sameer Ali, Muhammad Suleman Tahir

**Affiliations:** 1Faculty of Materials and Chemical Engineering, GIK Institute of Engineering Sciences and Technology, Topi 23460, Khyber Pakhtunkhwa, Pakistan; muzammil@giki.edu.pk; 2Interdisciplinary Research Center for Hydrogen & Energy Storage (IRC-HES), King Fahd University of Petroleum & Minerals, Dhahran 31261, Saudi Arabia; meali@kfupm.edu.sa; 3Department of Mechanical Engineering, Khwaja Fareed University of Engineering & Information Technology (KFUEIT), Rahim Yar Khan 64200, Punjab, Pakistan; kabeer.raza@kfueit.edu.pk; 4Mechanical Engineering Department, King Fahd University of Petroleum and Minerals (KFUPM), Dhahran 31261, Saudi Arabia; turkibaroud@kfupm.edu.sa; 5Department of Chemical Engineering, University of Gujrat, Gujrat 50700, Punjab, Pakistan; sameeralihakeem1@gmail.com (S.A.); vc@kfueit.edu.pk (M.S.T.); 6Department of Chemical Engineering, Khwaja Fareed University of Engineering & Information Technology (KFUEIT), Rahim Yar Khan 64200, Punjab, Pakistan

**Keywords:** polysulphone, carbon nanotube, graphene platelet, thermal properties, nanocomposite

## Abstract

Polysulphone (PSU) composites with carbon nanotubes (PSU-CNT) and graphene nanoplatelets (PSU-GNP) were developed through the solution casting process, using various weight load percentages of 1, 3, 5, and 10 wt% of CNT and GNP nanofillers. The microstructural and thermal properties of the PSU-based composites were compared. The microstructural characterisation of both composites (PSU-CNTs and PSU-GNPs) showed a strong matrix–filler interfacial interaction and uniform dispersion of CNTs and GNPs in the PSU matrix. The analysis demonstrated that both the thermal conductivity and effusivity improved with the increase in the weight percentage (wt%) of CNTs and GNPs because of the percolation effect. The polysulphone-based composite containing 10 wt% CNTs showed a remarkably high thermal conductivity value of 1.13 (W/m·K), which is 163% times higher than pure PSU. While the glass transition temperature (T*g*) was shifted to a higher temperature, the thermal expansion was reduced in all the PSU-CNT and PSU-GNP composites. Interestingly, the CNTs allowed homogeneous distribution and a reasonably good interfacial network of interaction with the PSU matrix, leading to better microstructural characteristics and thermal properties than those of the PSU-GNP composites. The findings highlight the importance of controlling the nature, distribution, and content of fillers within the polymeric matrix.

## 1. Introduction

Polymer materials have several advantages over metallic materials, such as high corrosion resistance, light weight, low cost, ease of processing, and chemical stability. However, there are obstacles in overcoming the high thermal expansion, low thermal conductivity, poor mechanical strength, and lower service temperature range [1]. The mismatch in thermal expansion between metallic and polymeric components is the primary cause of failure in polymer–metal joints and adhered surfaces [2]. Neat polymers usually exhibit thermal conductivity in the range of 0.1 to 0.4 W/m·K [3]. In contrast, most metals possess high thermal conductivity (100–400 W/m·K) [4].

Nanocomposites are multiphase solid materials and have one phase that is less than 100 nm in size. Nanosized particles with a high aspect ratio allow strong particle–particle interactions and lead to efficient heat energy flow. However, the ultrahigh interfacial area of the nanosized fillers permits a strong matrix-to-filler interaction at the level of the interface with the polymer matrix for the efficient transmission of thermal energy. This further strengthens the mechanical and thermal properties of the material [5]. Polysulphone is an amorphous polymer with a maximum continuous use temperature of 190 °C, and it can maintain its mechanical properties for an extended period of time [6]. It is utilised in membranes because of its good thermal stability and strength [7]. Polysulphone (PSU) is typically used in food processing equipment, bioreactor membranes, fuel cell membranes, and aviation and electronic components [8]. The polysulphone–hydroxyapatite (PSU+HAp) composite is biostable and can be utilised for orthopaedic applications [9]. Polymer nanocomposites filled with thermally conductive fillers such as aluminium nitride [10], CNTs, and graphene can be used to encapsulate electronic devices (photovoltaic systems) to protect them from adverse environments (i.e., high temperatures or corrosive), improve their performance, and extend their reliability [11,12,13,14,15,16]. Considerable research is being conducted to develop polymer nanocomposites with improved thermal properties and lifetime behaviour in the heating, ventilation, air conditioning, and refrigeration (HVAC and R) environment; computer chips; electronic packaging; LED systems, and many other conventional and advanced applications [17,18,19] Thermal conductivity is a bulk property of polymer nanocomposites. The principal factors that affect thermal conductivity are the filler concentration, the thermal conductivity of the filler, particle size, and interfacial thermal resistance [20]. Superior thermal properties can be obtained using suitable metallic, nitride, and oxide fillers such as aluminium nitride, cubic boron nitride, hexagonal boron nitride [21], strontium ferrite, magnetite, barite, and copper [22]. TiO_2_ addition (1–2 wt%) in PSU provides high thermal stability, and 40 vol% hexagonal boron nitride provides ideal thermal stability with thermal conductivity of 1.1 W/m·K [23]. The thermal conductivity increases from 0.22 to 0.93 W/m·K upon the addition of 44 vol% of magnetite (Fe_3_O_4_) [24]. Zhao et al. produced hybrid composites of graphene foam with carbon fibre (CF)-reinforced polydimethylsiloxane (PDMS) via solution mixing and high-speed shear mixing techniques. They showed that PDMS filled with 10 wt% CF infiltrated into graphene foam could provide a bulk thermal conductivity of 0.55 W/m·K. The effect of carbon fibre on the thermal properties of the graphene polymer composite showed that with 10 wt% CF, the maximum thermal conductivity reached was 0.55 W/m·K [25]. Yang et al. developed epoxy-based hybrid nanocomposites by introducing CNTs and GNPs through a solution casting route. The mixing of high- and low-aspect-ratio fillers resulted in a synergistic enhancement of the thermal and mechanical properties [26]. Similarly, a synergistic improvement in thermal conductivity of 0.86 W/m·K with the addition of 10 wt% CNT with Al_2_O_3_ in the PDMS matrix has been reported [27].

A high coefficient of thermal expansion (CTE) is a limiting factor in the utilisation of polymers. The most common thermoplastics used in polymer composite industries have thermal expansion in the range of 25–200 ppm/K [28], which can be adjusted to lower expansion values by incorporating fillers with a negative (or lower) coefficient of thermal expansion. It is possible to attain a material with approximately zero thermal expansion combined with positive and negative expanding materials for various applications [29]. Graphene and CNTs at room temperature (25 °C) have a negative coefficient of thermal expansion, so they can be added to polymer matrices to lower the thermal expansion for polymer composites effectively; however, the coefficient of thermal expansion is not affected by the addition of less than 0.5 wt% of CNTs and GNP [30]. Conductive polymer composites composed of carbon nanotubes and graphene have been examined for use in strain-sensing devices [31].

At room temperature (25 °C), the observed thermal conductivity of MWNTs is more than 3000 W/m·K and that of single-layer graphene is approximately 5000 W/m·K [32], owing to its very high phonon mean free path [33]. Carbon nanotube (CNT) addition in a polymer matrix leads to improved percolation at lower loading and such composites have a good estimation of the thermal properties using different models [34]. However, a higher filler loading on the nanoscale is likely to agglomerate and reduce the aspect ratio and effective surface-area-to-volume ratio [35]. The number of interaction points (between the fillers and the polymer matrix) in the percolation network was enhanced by the addition of nanosized fillers [36]. There are issues related to the effective dispersion of CNTs in the polymer matrix—that is, accumulation, the limited availability of high-quality CNTs, and their production cost [37]. Therefore, graphene sheets offer another option for producing nanocomposites filled with graphene because of their abundant graphite precursors [38]. Graphene with ultra-high thermal conductivity is a promising material for distributing heat generated in microelectronics [39]; with shallow filler content, it can significantly improve the thermal properties [40].

The thermal properties of similar filler materials vary drastically in various polymer matrices, and a comparison of multiple materials is difficult [24]. For many polymer composite materials, further research is required to probe the essential thermal property data, including the thermal conductivity, maximum operating temperature, and coefficient of thermal expansion. Nanofillers can enhance the ultra-low thermal conductivities of polymers with a super-high thermal conductivity of approximately 1000 W/m·K at room temperature (25 °C) [41].

In this study, GNPs and CNTs were systematically added into the PSU matrix via a solution mixing process to design polymer nanocomposites with improved thermal properties. The precursors and final composite materials were characterised in terms of X-ray diffraction, microstructures, Raman scattering, thermal conductivity, thermal expansion, and the glass transition temperature.

## 2. Materials and Methods

The matrix PSU was procured from Sigma-Aldrich (USA), the CNTs from Cheap Tubes (USA), and GNPs from Graphene Supermarket (USA). The GNPs were composed of 30 to 50 monolayers, with average thickness in the range of 12 nm and an average lateral size of ~3 µm. The matrix and reinforcement compositions in wt% are shown in Table 1, along with their digital illustrations, along with Table 1.

The nanocomposites were processed using the solution casting method [42]. Initially, specific amounts of CNTs and GNPs were added to 10 mL of N-Methyl-2-pyrrolidone (NMP). The mixture was sonicated for 2 h using a probe sonicator (Model VC 750, Sonics, Newtown, CT, USA) to obtain a stable suspension of the different compositions as indicated in Table 1. After the preparation of the solution, the required amount of PSU was dissolved in 10 mL of NMP in a separate container by magnetic stirring for a further 2 h. The mixtures were then mixed and stirred over a magnetic hot plate for another 12 h. N-Methyl-2-pyrrolidone (NMP) was used as a solvent to dissolve PSU into a suspension with different weight percentages (0, 1, 3, 5, and 10 wt%) of reinforcements (CNTs/GNPs). The stirring of the mixture was continued for 12 h using a magnetic stirrer to obtain a fully homogenised mixture.

The homogenised solution was cast in a petri dish and kept in a vacuum oven for 24 h at 80 °C to evaporate the solvent completely. Upon drying, the developed nanocomposite film was removed for necessary characterisation and evaluation of the thermal properties. The thermal conductivity and effusivity were measured at room temperature, 25 °C, using thermal analyser equipment C-Therm-TCi, (Fredericton, NB, Canada), for which the measurements were based on the modified transient plane source method. The coefficients of thermal expansion (α) were measured using a Mettler Toledo thermo-mechanical analyser (TMA/SDTA-LF/1100, Greifensee, Switzerland) from room temperature (25 °C) up to 180 °C at a heating rate of 283 K/min (10 °C/min). The T*g* was evaluated as a function of the filler composition using differential scanning calorimetry (DSC) (Mettler Toledo, Greifensee, Switzerland).

The phase analysis of the composites was conducted using X-ray diffraction experiments performed using a Benchtop MiniFlex X-ray diffractometer (Rigaku, Tokyo, Japan). The XRD was operated by maintaining a tube current of 10 mA, an accelerating voltage of 30 kV, and Cu Kα1 radiation. A Raman microscope (Thermo Scientific, DXR2, Boston, MA, USA) was used to observe molecular disorientation and band characteristics with an excitation wavelength of 455 nm (laser power of 2.5 mW), and the spectra were acquired at 25 °C between 200 and 3500 cm^−1^. A field emission scanning electron microscope (FESEM) from Tescan (Lyra3, Tescan, Brno, Czech Republic) was used to characterise the surface morphology of the composites at an accelerating voltage of 20 kV.

## 3. Results and Discussion

### 3.1. Microstructural Characterisation

Figure 1 presents the FESEM images of the as-received CNTs and GNPs. Figure 1a shows that as-received CNTs were entangled, constructing a percolating network in the polymer matrix. The entanglement of CNTs is viewed as a beneficial factor in constructing a stable conductive network in the polymer matrix. Figure 1b shows the stacks of multilayered graphene nanoplatelets with an average thickness of approximately 12 nm.

The XRD spectra of the pristine PSU and the composite samples are shown in Figure 2. The XRD pattern of pristine PSU is free of any sharp crystalline peak, which indicates its amorphous nature and typical characteristics of transparent and thermoplastic polymers. No crystalline peaks appeared in samples S1-S4 upon adding 5 wt% of PSU filler into CNTs and GNPs (Table 1). Samples S8 and S9 showed sharp graphite peaks at 2θ = 26°. However, the corresponding samples with CNT addition (S4 and S5) did not display any sharp peaks at the same diffraction angle. This observation is attributed to the more significant molecular weight of GNPs, which produced comparatively higher crystallinity in S8 and S9 samples.

Analysis of the XRD patterns of the pristine PSU and PSU-CNT and PSU-GNP nanocomposites (Figure 2) showed the crystallinity of the composites. The XRD pattern of pure PSU (Figure 2(1)) showed a broad diffraction peak at approximately 2θ = 18°, indicating a primarily amorphous structure for PSU. In the PSU-CNT nanocomposites, there was no indication of a sharp peak, even in the presence of CNTs in the polymer matrix. Notably, when the CNT content was increased to 10 wt% (Figure 2(5)), a diffraction peak appeared at approximately 2θ = 26°. For PSU-GNP nanocomposites, with the addition of a small amount of graphene (1–3 wt%), the nanocomposites showed identical XRD patterns to pure PSU. However, when the graphene content was increased to 5 or 10 wt%, a diffraction peak was observed at approximately 2θ = 27°, which demonstrates the aggregation of graphene nanoplatelets, especially for the 10 wt% graphene (Figure 2(9)). The absence of a peak in the composites with lower concentrations confirms dispersion within the reinforcements in the matrix [43]. Peak broadening at higher concentrations (Figure 2(5),(9)) demonstrates a slight change in the polysulphone structure, which perhaps may be due to the dominant interactions of weak Van der Waals forces.

The molecular disorientation and band characteristics were observed via Raman spectroscopy. The Raman spectrum of pure PSU shows typical peaks at ~790, ~1147, ~1583, and ~3065 cm^−1^. The band characteristics of PSU, CNT, and GNPS (composite) are shown in the Raman spectra in Figure 3, which shows that the D and G bands appeared at 1175 and 1650 cm^−1^, respectively. These are slightly shifted from the characteristic peaks (i.e., centred at 1345 and 1565 cm^−1^) of carbon nanoallotrope materials [44]. The increase in the intensity ratios of the D to G bands shows a reduction in the order (carbon atoms) as the amount of reinforcement increases. This phenomenon can be attributed to hydrogen bonding interactions between the polymer matrix chains and reinforcements [31].

The D band peak is associated with disorder arising from the Raman scattering process in disordered carbon atoms. In contrast, the G band peak results from the in-plane tangential stretching of carbon–carbon bonds. However, the relative intensity ratio of ID/IG is a good indicator of disorder; a higher value of this ratio indicates a higher number of defects [45]. Therefore, the defect level can be assessed by determining the intensity ratio of the D and G band peaks (ID/IG ratio) [36]. The ID/IG ratios of the 10 wt% CNTs and graphene polymer composites were 1.6 and 1.4, respectively, indicating that the defect level of the 10 wt% CNT composite was higher than that of the 10 wt% GNPs. The ID/IG ratio increased, confirming that the fillers were structurally intact and their concentration governed the Raman spectra of the developed nanocomposites [44]. Figure 4 shows the FESEM micrographs of the fractured surfaces of the PSU-CNT and PSU-GNP composites. The formation of a three-dimensional CNT network and interconnectivity in the PSU matrix can be observed in the FESEM images (Figure 4b–i).

The developed composites possessed a larger contact area and better interaction between the CNTs, as well as improved matrix percolation and synergistic enhancement. The interfacial interaction network formation in both types of reinforcements was excellent and suitable for a conductive path with high thermal conductivity. At higher loadings (10 wt%), the dispersion of CNTs is better than GNPs because intermolecular Van der Waals forces are much weaker in CNTs when compared with GNPs. It is evident from Figure 4f–i that the GNPs were well-dispersed in the matrix at lower concentrations, showing good compatibility when compared with higher concentrations. Hence, it was observed that the CNTs had better and easier dispersion in the PSU matrix than the GNPs.

The PSU-GNP composite with 10 wt% of GNP was closely combined and stacked, leading to the aggregation of graphene, as shown in Figure 4i, which was also confirmed by the XRD peak at 2θ = 27°. The strong surface–surface attraction between the filler sheets could lower the dispersion in the matrix [38]. GNPs have strong Van der Waals forces between the adjacent planes of carbon rings, which potentially results in agglomeration, defects, or flaws in the PSU-GNP composites (Figure 4i) when compared with the CNT counterpart. Therefore, well-dispersed, thermally conductive fillers play a vital role to amplify phonon transport and hence to synergistically enhance thermal conductivity.

### 3.2. Thermal Properties

The thermal conductivity and effusivity of the PSU-CNT and PSU-GNP composites that resulted from the incorporation of fillers are shown in Figure 5 and Figure 6, respectively. Whereas the setup for thermal conductivity and effusivity measurement was simply based on a sensor and processer, the setup for measuring the thermal properties used the modified transient plane source (MTPS) method. It requires only one sample to be placed on the disk-shaped sensor. The sensor also works as a heat source and simultaneously measures the transient effect of heat pulses to evaluate the thermal conductivity of the sample.

The experimental thermal conductivity results of the PSU-CNT and PSU-GNP composites were correlated with the theoretical model published by Raza et al. [46], which can be used for dilute concentrations of percolating single and hybrid fillers in polymer composites. It was observed that the results were close to the predictions of the model at lower concentrations of up to 5 wt%. However, the deviations at 10 wt% can be attributed to the agglomeration among the nanoparticles and related defects, which were ignored by the theoretical model.

The thermal conductivity of the PSU-CNT and PSU-GNP nanocomposites varied as a function of the weight percentage and type of filler (CNTs or GNPs). The heat conduction or transmission of thermal energy requires interparticle connections. A strong interfacial interaction of particles with a combination of vibrations (i.e., phonons) and free electrons supports the fast transport of heat carriers (phonons and electrons). Phonon transport is the dominant mechanism in polymers, carbon, and their composites [47]. The mean free path for phonons in the polymer is lower (a few angstroms) compared to the mean free path for CNTs (hundreds of nanometres). Thermoplastics typically exhibit lower thermal conductivity and phonon transport inhibition because of their lower density, structural inhomogeneity, and mismatch in molecular vibrations. More importantly, their interfacial thermal resistance decreases the thermal conductivity of polymer nanocomposites [48], which has been predicted in developed models [49]. The fundamental transport and scattering of heat carriers (phonons and electrons) also depend on the bulk properties, whereas the properties of interfaces and finite dimensionality are vital issues in their applications [14]. The PSU-CNT composite with 10 wt% CNTs showed increased thermal conductivity compared to that of the PSU-GNP nanocomposite with 10 wt%.

The polysulphone–CNT composite with 10 wt% CNTs shows an enhancement in thermal conductivity of 163%, which is approximately 50% higher (increase) in comparison to the addition of 10 wt% graphene nano-platelets. This can be ascribed to the formation of an interconnecting network with strong coupling at the PSU-CNT interface in the case of CNTs, as predicted from the FESEM micrographs (Figure 4b–e). The formation of the interconnecting network is much better at lower concentrations—that is, 1–5 wt% CNT/GNPs (Figure 4b,c,f,g). The formation of an interconnecting/percolating network promotes heat flow. It reduces the phonon scattering at the interface between the reinforcements and polymers, which can improve the thermal transport in composites [41]. At higher reinforcement loading, interfacial thermal resistance obstructs further improvement in the thermal transport of charge carriers (phonons and electrons). The thermal properties of composites are a function of the volume fraction, aspect ratio, alignment, and adhesion interface condition between the particles and the matrix [50]. Nanofillers (such as CNTs and GNPs) with a high aspect ratio form a percolating network. These are promising candidates for high conductivity owing to their high-aspect-ratio nanofillers [33]. However, for graphene, there is a high probability of sheet-to-sheet contact even at comparatively low graphene loading because of the overlapping electronic clouds of neighbouring carbon atoms [38]. However, these conditions are well shown in Figure 4f–i; with better distribution, the conductivity can be improved to 114% compared to that of a neat (pristine) polymer. Thermal effusivity is a measure of the ability of a material to exchange thermal energy with its surroundings [51]. Table 2 shows the positive correlations between the amount of filler and the thermal effusivity.

The measured coefficients of thermal expansion (CTE) of the samples are shown in Table 2 and Figure 7 as a function of the filler concentration.

Higher concentrations of CNTs and GNPs (10 wt%) led to much lower CTE values than pristine PSU. The CTE values were reduced by 36% and 24% with the addition of CNTs and GNPs, respectively. The reduction in CTE is related to the bonding forces between the atoms and the available free volume in the polymer matrix. With the increase in the filler concentration, the bonding forces of the composite structure increased, and the free volume decreased. Hence, the expansion caused by the temperature change was restrained, resulting in a lower CTE (which is higher at a 10 wt% loading, as shown in Figure 4a–i). Hence, the high concentration of fillers contracts the space available for the expansion of the polymer and ultimately reduces the CTE of the composites (Table 2). The second-order phase transition phenomenon, which is known as the T*g*, is common in polymeric and noncrystalline materials. The material transforms from a brittle, crystalline, semicrystalline solid into an elastic, amorphous solid [52]. The T*g* values of the pristine PSU and its composites are shown in Table 2 and Figure 8.

The pristine PSU exhibited a T*g* of 190 °C. After the addition of CNTs, the T*g* of the nanocomposites was shifted to a higher temperature, which showed the improved thermal stability (to a certain extent) of the composites. In this study, after the addition of the reinforcements, the T*g* increased from 190 °C (for pristine PSU) to 250 °C (with a 10 wt% addition of CNTs). The shift in the T*g* to a higher temperature can be attributed to the close affinity between the matrix and reinforcements owing to intermolecular hydrogen bonding at the interface [53]. The T*g* value increases with an increase in the weight fraction of fillers in a polymer matrix, which is credited to the improved compatibility and interaction between the matrix and the filler. However, it was observed that with the addition of either type of filler (CNTs or GNPs) from 5 wt% to 10 wt%, there was no significant increase in T*g*. The T*g* value depends on the free volume and interface connections between the fillers and the matrix. Therefore, compositions with better affinity to fillers show higher T*g* values owing to less molecular motion and the reduced free volume of the polymer molecules [27,28]. Better distribution and cross-linking of embedded reinforcements in the polymer matrix, as shown in Figure 4b–i, reduce the motions of polymer molecules, resulting in an increase in T*g* (Figure 8) and a reduction in the thermal expansion (Figure 7). In addition to this, another critical observation was that there was no significant improvement in the observed thermal properties from 5 to 10 wt% fillers (Table 2). Hence, based on the experimental observations, it is recommended that simple fabrication techniques with the correct selection and combination of fillers in the polymer matrix can aid in the design of polymer composites at an economical rate for various industrial applications.

## 4. Conclusions

Polysulphone (PSU) nanocomposites with carbon nanotubes (CNTs) and graphene nanoplatelets (GNPs) with the addition of 1–10% weight percentage were prepared using the solution casting technique. The FESEM micrographs showed reasonable dispersion and established solid interfacial bonding between the PSU and both types of nanofillers (CNTs and GNPs). Notably, the developed composites (PSU-CNT and PSU-GNP) showed improved thermal properties such as stability, conductivity, and expansion upon the addition of CNTs and GNPs from 1 to 10 wt%. The thermal conductivity and glass transition temperature (T*g*) of 10 wt% of PSU-CNT were 1.13 W/m·K and 250 °C, respectively, which were better than the 10% wt of PSU-GNPs (0.92 W/m·K and 240 °C), and there was an increase of approximately 163% compared to pure PSU. In comparison, the composite with 10 wt% CNTs in PSU possessed a more favourable combination of three-dimensional interconnected microstructure and thermal properties for thermal applications. These results highlight the significance of the type, composition, and distribution of the nanofiller in the microstructural and thermal characteristics of the polymer nanocomposites. Hence, this study paves the way towards the design and development of promising polymer composites with beneficial thermal properties to be used in niche applications.

## Figures and Tables

**Figure 1 nanomaterials-11-02080-f001:**
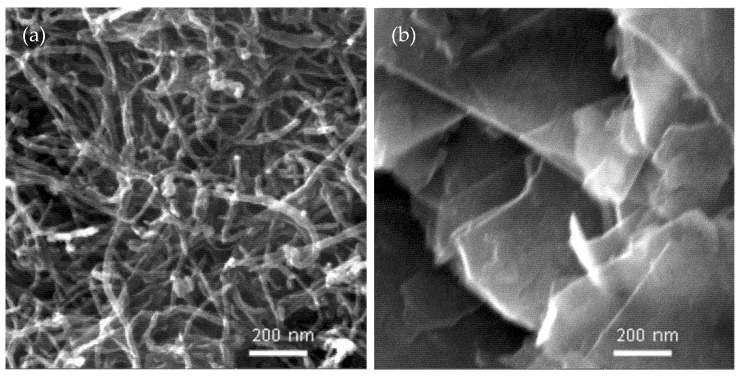
FESEM micrographs of pure CNTs (**a**) and GNPs (**b**).

**Figure 2 nanomaterials-11-02080-f002:**
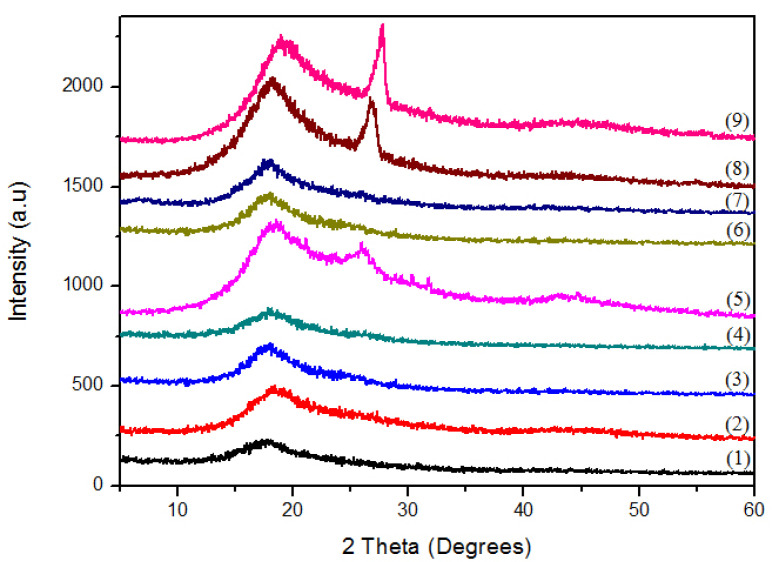
X-ray diffraction (XRD) patterns of PSU-CNT and PSU-GNP samples. Numbers (1)–(9) in parentheses correspond to numbers of samples for which compositions are described in Table 1.

**Figure 3 nanomaterials-11-02080-f003:**
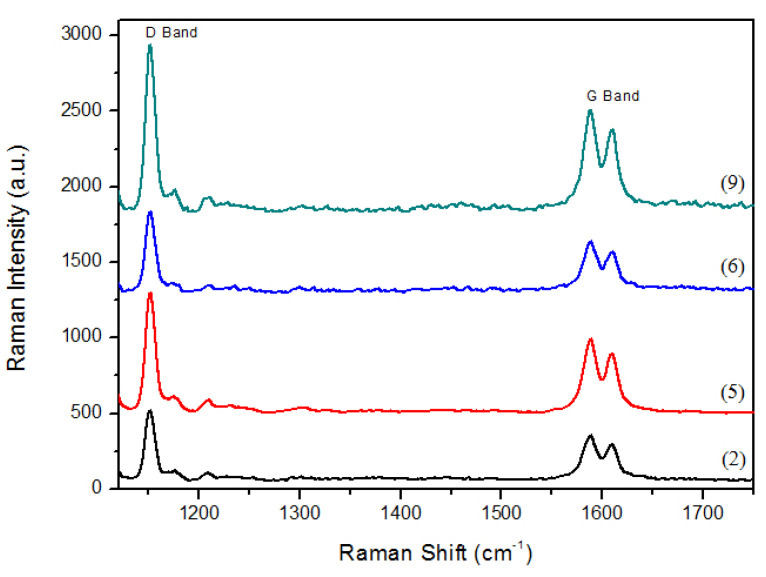
Raman analysis of PSU-CNT and PSU-GNP nanocomposites. Numbers (2), (5), (6), and (9) in parentheses correspond to sample IDs and compositions described in Table 1.

**Figure 4 nanomaterials-11-02080-f004:**
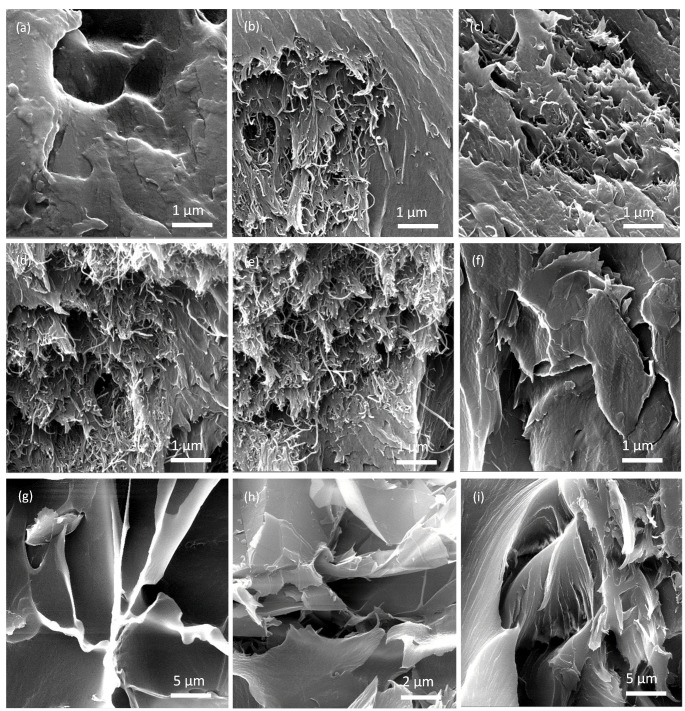
FESEM micrographs of compositions: (**a**) pure PSU, (**b**) PSU-1wt%CNT, (**c**) PSU-3wt%CNT (**d**) PSU-5wt%CNT, (**e**) PSU-10wt%CNT, (**f**) PSU-1wt%GNP, (**g**) PSU-3wt%GNP, (**h**) PSU-5wt%GNP, and (**i**) PSU-10wt%GNP.

**Figure 5 nanomaterials-11-02080-f005:**
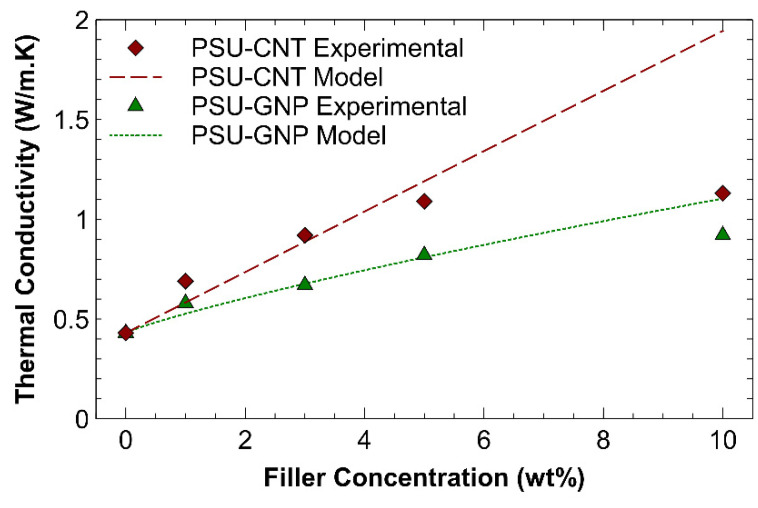
Effect of adding CNTs and GNPs on thermal conductivity of the resultant composites and their correlation with theoretical model.

**Figure 6 nanomaterials-11-02080-f006:**
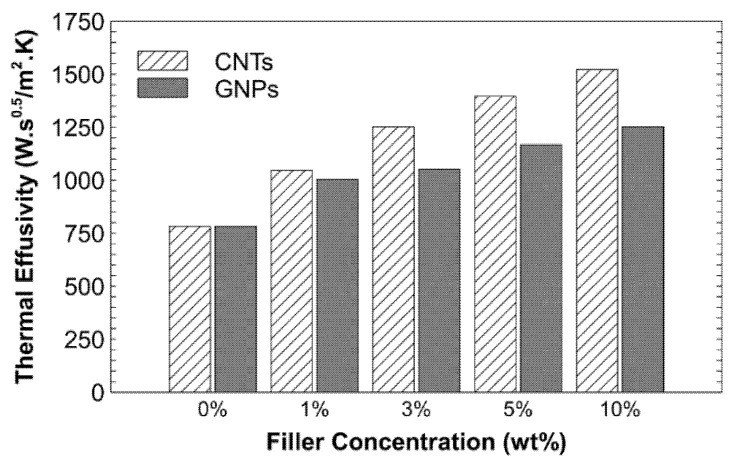
Effect of adding CNTs and GNPs on thermal effusivity of the resultant composites.

**Figure 7 nanomaterials-11-02080-f007:**
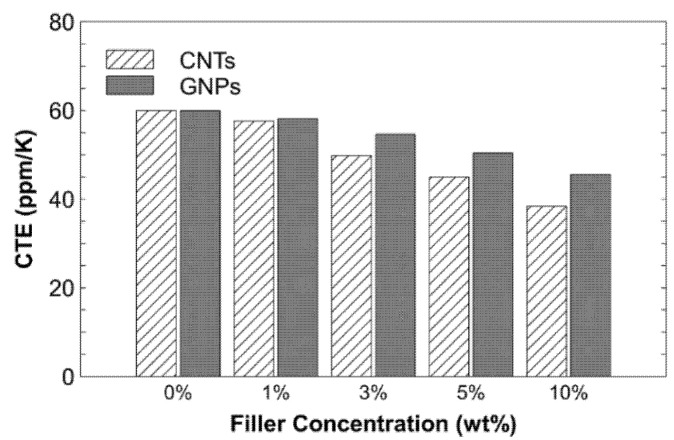
Effect of adding CNTs and GNPs on the coefficient of thermal expansion of the resultant composites.

**Figure 8 nanomaterials-11-02080-f008:**
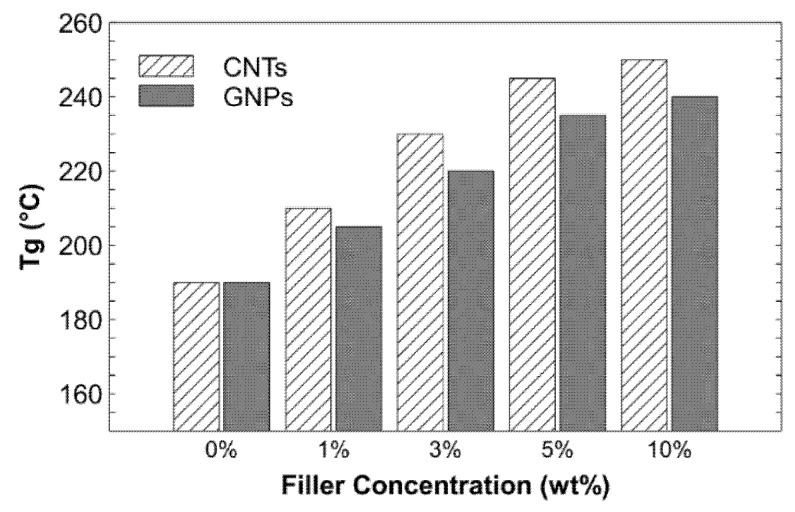
Effect of adding CNTs and GNPs on glass transition temperatures (T*g*) of composites.

**Table 1 nanomaterials-11-02080-t001:**
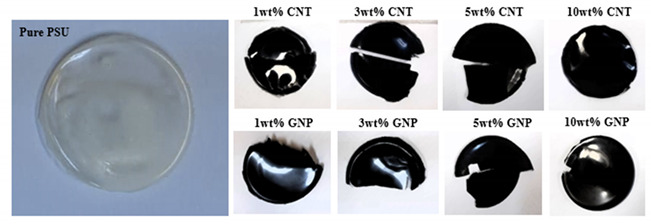
Compositional variation of PSU and reinforcements (CNT/GNP).

Composition
Sample IDs	1	2	3	4	5	6	7	8	9
Composition (%)	Pure PSU	1% CNT	3% CNT	5% CNT	10% CNT	1% GNP	3% GNP	5% GNP	10% GNP
PSU (gm)	10	9.9	9.7	9.6	9	9.9	9.7	9.6	9
CNTs/GNPs (gm)	0	0.1	0.3	0.4	1	0.1	0.3	0.4	1

**Table 2 nanomaterials-11-02080-t002:** Thermal properties of PSU-CNT and PSU-GNP composite samples; the sample IDs are according to Table 1.

Composition									
Sample ID	1	2	3	4	5	6	7	8	9
Composition	Pure PSU	1% CNT	3% CNT	5% CNT	10% CNT	1% GNP	3% GNP	5% GNP	10% GNP
Thermal Conductivity (W/m·K)	0.43(3)	0.69(3)	0.92(5)	1.09(1)	1.13(7)	0.58(6)	0.67(3)	0.82(2)	0.92(3)
% Increase in Thermal Conductivity	0	60	114	153	163	35	56	91	114
Thermal Effusivity	782(3)	1047(7)	1252(6)	1397(3)	1522(2)	1005(3)	1052(5)	1167(6)	1252(8)
Thermal Expansion (ppm/K)	60(4)	57.6(6)	49.8(3)	45(5)	38.4(1)	58.2(2)	54.6(3)	50.4(1)	45.6(5)
% Reduction of Thermal Expansion	0	4	17	25	36	3	9	16	24
T*g* (°C)	190(2)	210(2)	230(6)	245(8)	250(7)	205(4)	220(3)	235(4)	240(2)

## Data Availability

Not applicable.

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
