# Peer review of "Design, Development and Evaluation of Thermal Properties of Polysulphone–CNT/GNP Nanocomposites"

_nanomaterials, 2021, doi:10.3390/nano11082080_

Round 1
Reviewer 1 Report
should be more clarified on the presentation of dispersion and aggregation of fillers (lines 264 - 276) (may be authors add a table showing the comparison between the fillers?).
Author Response
First, the authors would like to appreciate the Editor and the Reviewers for making necessary comments and suggestions regarding the subject paper. The required changes and corrections are incorporated in the revised manuscript and highlighted in yellow in the manuscript.
Response to the reviewer comments are as follows, highlighted in red colour text.
Response to Reviewer 1 Comments
Point 1: Should be more clarified on the presentation of dispersion and aggregation of fillers (lines 264 – 276) (maybe authors add a table showing the comparison between the fillers?).
Response 1:
The text in lines 270-283 has been updated as guided and advised by the reviewer.
Thank you

Reviewer 2 Report
Please see the attachment.

Author Response
First, the authors would like to appreciate the Editor and the Reviewers for making necessary comments and suggestions regarding the subject paper. The required changes and corrections are incorporated in the revised manuscript and highlighted in yellow in the manuscript.
Response to the reviewer comments are as follows, highlighted in red colour text.
Response to Reviewer 2 Comments
Point 1: An interesting study concerning experimental investigations of thermal conductivity, effusivity, stability, and thermal expansion of hybrid nanocomposite is reported.
Response 1:
We appreciate and are grateful for the reviewer’s positive comments.
Point 2: The results obtained are new, and valuable, and potentially important from practical point of view.
Response 2:
We appreciate the reviewer’s positive comment.
Point 3: The paper is well written, clearly readable.
Response 3:
We appreciate the reviewer’s positive comment.
Point 4: Presented results are reliable and well documented.
Response 4:
We appreciate the reviewer’s positive comment.
Point 5: Some issues should be clarified:
- Replace Fig. 5 and Fig 6.
- Line 152: instead of °C/min. should be K/min.
- 7. The unit of CTE – (ppm/K) may be confusing for some readers.
Response 5:
- Figures 5 and 6 are replaced to correct positions as suggested.
- In Line 152, °C/min is replaced to K/min, as suggested
- We appreciate the reviewer’s positive comment. However, to make units consistent throughout the manuscript, the authors used the same units (CTE – (ppm/K = micron/K) in figure 7.
Point 6: Editorial Please, correct the titles of the journals: [11], [13], [16], [27], [32], [33], [40], [41], [44], [47], [50].
Response 6:
Corrected as suggested, thank you!

Reviewer 3 Report
The authors report on the “Design, development and evaluation of thermal properties of Polysulfone-CNTs/GNPs nanocomposites”.
The work is in general well-structured and the paper is very well-written.
Moreover, a great number of techniques have been included in order to support the main findings of this work.
The work is quite innovative for the field of smart and multi-functional polymer nanocomposites with many potential applications, making use of advanced carbon allotrope nanomaterials
Some minor comments that should be considered:
- Abstract: Please mention the method for preparing the nanocomposites (i.e. solution mixing)
- Materials and Methods section: please divide this section into different paragraphs. This helps the better reader’s comprehension.
- Underneath Table 1, can you please insert digital photos with the different samples. It will be nice for the reader to get an idea how the films look like.
- Can you measure electrical conductivity as well. It is very interesting to see the inter-correlation of electrical and thermal transport in the fabricated nanocomposites, as well as the percolation of electrical and thermal conductivity (if they match or not)
- Field emission SEM at 20kV. Are you sure? Also can you show the samples that have been used for the SEM analysis (films, powder on some sample holder substrate or?). In principle the microscopy images in Figure 1 do not provide and scientific information.
- For the Raman analysis, please consider polymer thermoplastic/ Carbon nanofiller nanocomposite analyses (i. Polymer Volume 131, 22 November 2017, Pages 1-9; ii. C 2021, 7(2), 38) to give a better discussion about the polymer fingerprints as well as the carbon nanoallotrope characteristic peaks. The Raman discussion should be improved.
- It will be nice of the authors could provide some steady state and transient infrared thermography images to demonstrate the improvements in the thermal conductivity for the different nanocomposites.
- It will be really educative the authors to include some thermal analysis results e.g. DSC and TGA if possible.
- Finally, including a schematic showing the principle of the thermal conductivity, some main equations and the cartoon of the set-up and its operation (or even a real photo of the set-up) will be really helpful for the better reader’s comprehension. Thermal conductivity is the result of phonon transport as well as the thermal energy that is transferred through the electronic carriers (k = kph + kel). Is it possible that the authors will sure some known models to calculate the thermal conductivity and correlate it with the experimental findings. This will improve significantly the study (at scientific level) and also it will make the work much more complete.
In terms of originality, importance & scientific quality, relevance & contribution to the field and presentation, this manuscript is of good level.
Furthermore, the discussion in different paragraphs could be improved in the points that have been indicated in order to make the manuscript more interesting to the reader and more educative.
The manuscript and its content are sufficiently novel to warrant its publication, however, after including and considering the additions and clarifications proposed.
